# A Novel Approach to Identify Industrial Logistics Decarbonization Opportunities: Method Development and Preliminary Validation

Philipp Miklautsch-Breznik *, Mario Hoffelner and Manuel Woschank

Chair of Industrial Logistics, Montanuniversitaet Leoben, 8700 Leoben, Austria
* Correspondence: philipp.milkautsch@unileoben.ac.at

**Featured Application: This approach can assist industrial logistics professionals in obtaining a fresh viewpoint toward their decarbonization strategy.**

**Abstract:** This article explores how different types of inventories affect the costs of decarbonizing transportation in manufacturing companies. For these companies, it is difficult to find affordable ways to reduce emissions from transportation given their resource scarcity. Additionally, they handle numerous inventory items that have varying transportation needs based on their order frequency and value, which necessitates the development of tailored inventory management strategies. One tool to do so efficiently is the ABC/XYZ analysis, which classifies items into nine different inventory categories. These groups have different economic importance and predictability, which impacts total logistics costs. Our literature analysis contends that lower-carbon transportation alternatives yield varying abatement costs contingent upon the specific inventory categories. Subsequently, we empirically validate this proposition through discrete-event simulations in two case studies involving Austrian manufacturing enterprises, employing combined road-rail transportation as an illustrative decarbonization measure. Statistical tests substantiate the significance of the XYZ dimension in influencing carbon emission abatement costs during the transition from road to rail transportation. In conclusion, our study offers a novel perspective on decarbonization efforts, underscoring the importance of leveraging established management tools to inform strategic decarbonization decisions. This research holds promise for catalyzing progress in overcoming entrenched challenges associated with decarbonization initiatives within industrial logistics.

**Keywords:** green logistics; decarbonization; climate change; industrial logistics; ABC analysis; XYZ analysis; simulation; transportation

## 1. Introduction

Decarbonizing logistics is, from a scientific perspective, crucial for mitigating climate change [1], and from an institutional perspective, crucial for achieving net-zero pledges [2]. Systematic approaches to developing strategies for logistics decarbonization from a freight owners' perspective already exist. For example, the 10C approach by McKinnon [3] consists of ten activities and includes possible options to consider. The field of developing carbon mitigation measures for transportation is well researched, from discussing different drivetrain technologies [4] and alternative fuels [5] in technical terms to developing green routing algorithms [6]. Regarding the application of greening measures in practice, generic total cost of ownership studies exist, but often only distinguish between factors that are not tangible from the perspective of the freight owners, e.g., the application scenarios in urban logistics [7], factors on the national level [8], or barriers for the implementation of specific measures [9,10]. Although all these contributions are important for the decarbonization progress, efficient and effective guidelines that freight owners can use to select decarbonization measures are missing. Most studies only rank the different measures by abatement

costs. Although this is a common instrument to evaluate the efficiency of decarbonization measures and is highly important in selecting decarbonization measures [11–13], abatement costs only provide limited support for firms in identifying appropriate alternatives that can be ranked and prioritized based on their respective situation.

The challenge is to select a decarbonization measure that minimizes expenses and avoids disrupting operations [14,15]. This problem is particularly difficult to overcome when dealing with diverse goods that have varying value and volume compositions, which is the case in most industrial companies. To the best of our knowledge, there is currently no way for industrial companies to increase the effectiveness and efficiency of the process of identifying and selecting transport decarbonization measures for further evaluation.

Nevertheless, one positive result is that parallels can be drawn with another area of industrial engineering and management, that is, inventory management, which deals with many items and must prioritize them to manage inventory levels [16]. Therefore, instead of investigating each item on its own, a common approach is to classify goods with similar characteristics or importance to the firm's success. Inventory control tools for these purposes have been in use for economic reasons for many years. One of the best-known inventory control tools is the ABC/XYZ analysis, which classifies inventory items along the two dimensions, "value" and "order frequency," into nine types. For each of these types, different replenishment strategies are commonly applied to efficiently and effectively manage inventory levels [17].

With regards to these parallels, in this study, we introduce a novel approach to identify industrial logistics decarbonization measures that differentiate the transportation requirements of the transported goods in line with the ABC/XYZ analysis. Simply put, we propose that different transportation requirements come with different costs for lower-carbon transportation. From the inventory management perspective, higher transportation costs can be borne for some inventory items, as their storage is as expensive as their transportation [16]. If shipments in which such goods are transported are to be decarbonized, high abatement costs may have to be expected. For shipments of lower value or shipments with a stable demand, cost-effective decarbonization measures might be an option, leading to negative abatement costs. In the first part of this paper, we delineate this novel perspective from literature and define our proposition.

In the second part, we present the results of two simulation case studies, exemplarily testing the proposition using a promising decarbonization option, i.e., the shift from road to rail. We develop a discrete-event simulation using Python that mirrors the behavior of a logistics system using combined road–rail transportation. We decided to use this measure for several reasons, which are elaborated on in Section 4.1.

To summarize, the overarching goal of this paper is to introduce a novel perspective on carbon management for industrial logistics and provide the first evidence for its effectiveness. For researchers, this adds fertile ground for further research on the entrenched field of applying decarbonization measures in industrial logistics. As for practitioners, this paper offers an efficient way of investigating decarbonization measures to reduce their logistics' impact on climate change. Chances are high that ABC/XYZ analysis has already been implemented in a manufacturing company, which is why this approach proves useful for efficiently evaluating alternative and environmentally friendly transportation methods.

The paper is structured as follows: Section 2 presents the research methodology applied in this paper. Section 3 presents the results of a literature review on the selection of logistics decarbonization measures from an industrial company's perspective, the parallels to inventory management and the developed theory. Section 4 presents the argumentation for using combined road–rail transportation to evaluate the proposition and the method to do so. Section 5 outlines the case studies for the exemplary validation, the assumptions, and scenarios, as well as the simulation results. The paper concludes with a discussion of the limitations and implications in Section 6 and a summary of the results in Section 7.

## 2. Research Methodology

As the title of this paper suggests, this research is split into two consecutive parts. In the first part, the proposition is formulated. The initial motivation to do so emerged from the preceding research of the authors and discussion on decarbonization practices with industry experts. Throughout these talks, the authors recognized a repeating pattern where logistics managers stated that a specific decarbonization measure is only competitive for a certain type of good, i.e., a small portion of the inventory. Starting with this observation, we have gradually developed our theory with findings from the literature in a back-and-forth manner between literature and practice; we present it in the first part of this paper. This method of theory development is suggested to build theory from case studies [18] and applied by other researchers in supply chain management [19].

The second part of the paper deals with the validation of the developed proposition. However, since this proposition is very generic, it is not possible to test it in all its facets. For this reason, we have chosen a specific decarbonization measure to test the proposition. This measure is the shift from road transport to combined transport. We quantify the carbon abatement costs through a discrete-event simulation using primary data from two Austrian industrial companies.

The research process is visualized in Figure 1.

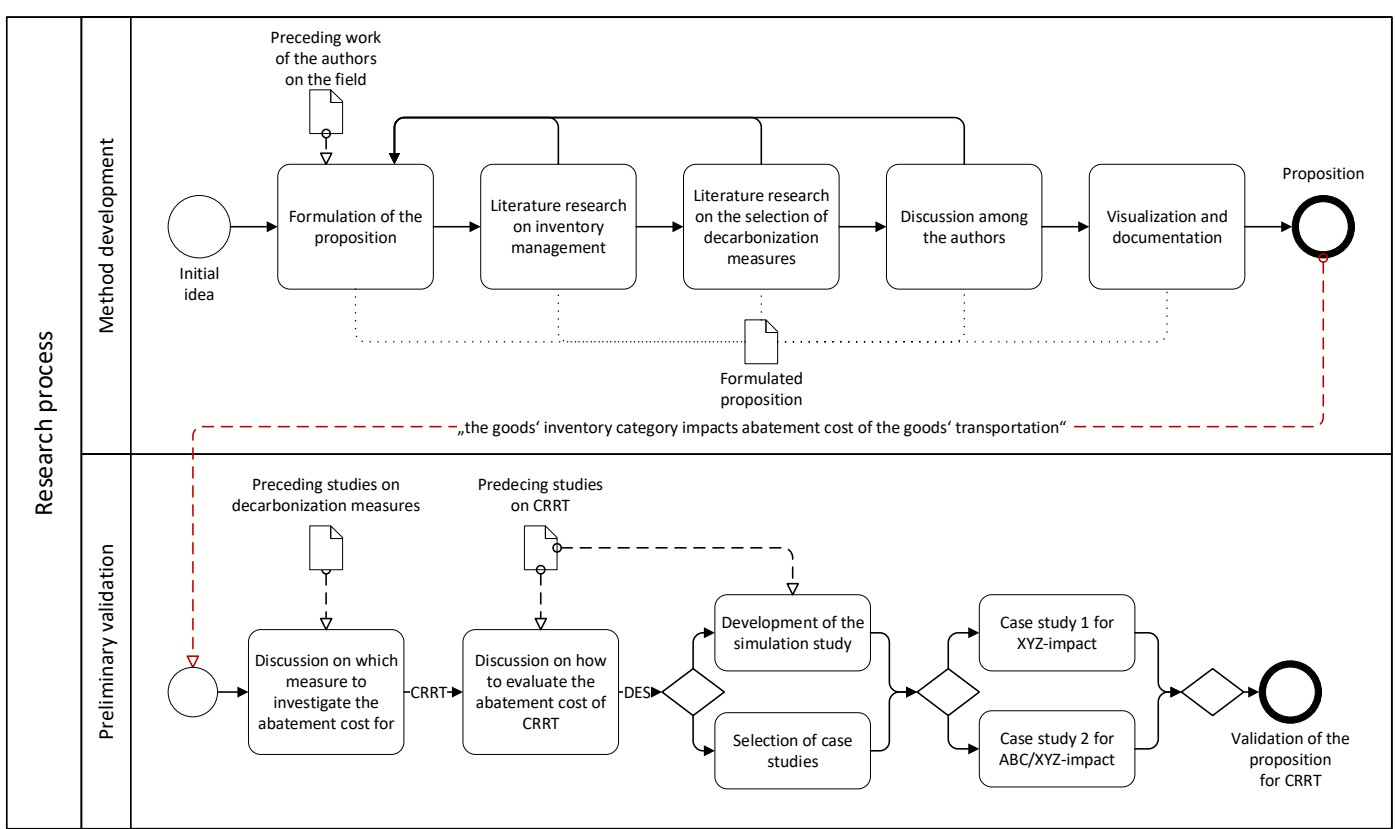

**Figure 1.** The research process of this study.

## 3. Theory Development

This section explains the challenges involved in selecting decarbonization measures from a manufacturing company's perspective and delineates our theory.

### 3.1. Selection of Decarbonization Measures

To decarbonize industrial logistics, researchers and practitioners discuss a vast number of different measures [20], which are commonly classified through the Avoid–Shift–Improve (ASI) approach [1,21,22]. As modern economic theories state [23], implementing all the

measures is impossible in practice due to resource scarcity. As economic entities only have limited resources available, policymakers and managers who define decarbonization pathways need to prioritize decarbonization measures [24] for some criteria. On a macroeconomic level, policymakers may prioritize the highest-emitting economic sectors in their decision scope [25]. Delving deeper into the sectors, such as the building sector, scholars suggest prioritizing actions according to their decarbonization potential, respecting their detailed technical feasibility [26]. Other researchers promote prioritizing those technologies that are least invasive to the current environment, i.e., rooftop photovoltaics in combination with electric vehicles [27,28].

Although these prioritization efforts are all, from an environmentalist perspective, comprehensible, actual decarbonization decisions in the industry are rarely one-dimensional. For example, when investigating decisions on a microeconomic level, empirical research found that decarbonization alternatives in the building sector were evaluated and prioritized by 27 different criteria [29]. Similar research approaches are applied to other sectors, including energy generation. In one case, green hydrogen production alternatives were prioritized according to their efficiency and sustainability, considering the criteria capital cost, feedstock cost, operations and maintenance cost, hydrogen production, and $CO_2$ emission [30]. These criteria focus on cost, which already implies that prioritization based solely on the effectiveness of decarbonization is unlikely. Instead, decarbonization may only be possible economically, which is a theory supported by various authors (for instance, see [31,32]), not least for freight transport. Exemplarily, the factors hindering the adoption of electric trucks in the United States were researched, and the results indicated that the top causal factors are the business model and partnerships, product availability, and charging time [9]. A Delphi Study on factors affecting the adoption of alternative fuel-powered trucks in Germany found that cost and reliability factors are ranked highest among practitioners [10]. These criteria, in the end, all reflect the efficiency of electric trucks. Similarly, the chief impediments mentioned by industrial logistics experts when surveyed about green practices were mostly related to costs [14]. A study proposing a multi-criteria decision-making tool for industrial logistics practitioners defined eight criteria for evaluating decarbonization alternatives. The two criteria considered most relevant in the demonstration case study were "abatement cost" and "impact on logistics performance" [15].

In summary, the selection of decarbonization measures in industrial logistics practice prioritizes efficiency over effectiveness. As previously demonstrated in [15], the economic efficiency of decarbonization measures can thereby be well expressed in terms of the abatement costs *AC*. These are defined by:

$$AC_m = \frac{C_m - C_b}{G_m - G_b}, [C] = \text{EUR}, [G] = t\,CO_2\,e$$

describing the cost difference of a decarbonization measure *m* compared to a baseline situation *b*, concerning the greenhouse gas (GHG) emissions reduced by *m*. Simply put, *AC* indicates how much it costs to reduce one ton of GHG and, thus, to which extent that measure is competitive with other measures. Abatement costs are commonly reported in studies dealing with decarbonization pathways or options, for example, in the energy sector [11,12]. Nevertheless, *AC* starts to be frequently used in freight transport, as well. In a recent study by Chinese researchers, various decarbonization measures for sand and gravel transportation were analyzed using the ASI strategy. By calculating *AC*, it was found that switching to lower carbon modes of transport was the most competitive option [22]. The cross-sectoral study of Denmark's transition to fossil-free transportation revealed differences in abatement costs both within and between transport segments [13]. An analysis of American electric vehicle procurement incentives revealed that the utilization of electric vehicles influences the abatement costs related to them [33].

According to these studies, the efficiency of decarbonization measures, in terms of abatement costs, varies significantly between implementation scenarios characterized by,

for example, weight, volume, frequency, origin, and destination. This emphasizes the challenge of identifying efficient decarbonization measures in the logistic network of a manufacturing company, which usually involves goods of varying priorities, suppliers, customers, and values. To the best of the authors' knowledge, it is still unknown how industrial firms, whose main competence lies outside the decarbonization of logistics, can identify efficient decarbonization measures effectively.

*3.2. Inventory Management*

Handling the above-mentioned challenge regarding the multiplicity of goods purchased, transported, and handled in a manufacturing company, is the responsibility of inventory management. Inventories frequently constitute a considerable portion of the total assets on the balance sheet of manufacturing companies; a 15–20% share is not uncommon [16]. Given that inventory holding costs can range up to 26%, inventories have a significant impact on a company's cost structure. The DuPont scheme incorporates inventories in the evaluation of the current asset in the return on investment (ROI) calculation and highlights the lever of inventory reductions: a 10% inventory reduction results in a 3.6% ROI increase [16].

Nevertheless, inventory management has been found to significantly affect both the economic success and sustainability of the company. Choices made on the inventory management level thereby influence a multitude of factors, e.g., the necessary distance, the frequency, or the mode of transportation. For instance, replenishment strategies establish the essential prerequisites for transportation processes in terms of frequency and volume, which, if optimized, can result in cost and environmental advantages [34]. Mode choice depends on lead time, volume, frequency, and costs [35]. Whereas fast modes enable priority shipments, they usually come with premiums. Slow modes enable efficient transportation of high volumes at one time and reduce the replenishment frequency but are less flexible [36]. Calibrating the inventory management strategy for the whole inventory is a complex task for a manufacturing company handling multiple products given the varying product requirements [37].

To effectively optimize inventory, organizations, therefore, commonly prioritize goods based on their relevance. One widely used tool for distinguishing highly relevant goods from less relevant ones is the ABC analysis. The analysis classifies articles, suppliers, or inventory movements according to their materiality for inventory management. The A category thereby comprises the highest-value elements that account for 70–80% of the value measured in monetary units. Usually, these elements only account for 5–10% of the number of elements, highlighting the most relevant ones for inventory management [38]. Regarding the inventory value, these elements are of utmost importance and provide the largest lever for improvements. Elements belonging to category B generally represent 25% in terms of quantity and 10–15% in terms of value. The rest of the elements, which are considered the least valuable, are categorized as C. These elements generally comprise 65% of the total quantity, but contribute only 10% to the overall value [38]. The ABC analysis is known for its ease of use and its clear graphical representation, making it a commonly used instrument among practitioners [16]. Nevertheless, it has been criticized frequently for its one-dimensionality, which is why researchers started to incorporate multi-criteria decision-making techniques in the field of inventory management [39]; they added, for example, non-financial criteria [40] and uncertainty [41] to the classical ABC analysis. One further dimension that has been used to detail the results of the ABC analysis is the demand fluctuation, which is investigated using the XYZ analysis [42]. Like ABC, XYZ classifies inventory elements into three categories but uses the coefficient of variation $CV$ instead of the element's monetary value. X elements, thereby, have a stable demand and a $CV < 0.1$; Y elements show seasonal fluctuations having $0.1 \leq CV < 0.25$; and Z elements are identified by $CV \geq 0.25$, being vastly volatile and unpredictable [16].

### 3.3. The Developed Theory

As elaborated above, inventory management and its ABC/XYZ analysis is a crucial and extensively researched aspect of operations management. Manufacturing companies typically use distinct logistics strategies for each of the nine emerging cells [35], which differ by the transportation mode, the order frequency, and the lead time [43]. These factors thereby impact the transportation cost, inventory holding cost, and order cost, which, together, form the total logistics cost [35,44,45]. Thus, the ABC/XYZ category of an inventory item impacts the logistics strategy applied to it, which, in turn, impacts total logistics cost.

Thereby, a goods' ABC dimension primarily affects the inventory holding cost, as it represents the pecuniary value of the inventory item. The XYZ dimension, on the other side, impacts the required lead times, flexibility, and resulting vehicle utilization, as it represents the demand characteristics, thereby impacting transport and order costs.

Combining this with the findings of Section 3.1, that abatement costs for one decarbonization measure differ among transportation scenarios, we can, in turn, delineate that logistics strategies impact the abatement costs of specific measures.

Thereby, the abatement costs represent the total logistics cost difference between the conventional transportation technique and a lower-carbon alternative, divided by the mitigated carbon emissions. As the goods' category impacts the total logistics cost, we theorize that the goods' category also impacts the abatement costs of reducing carbon emissions from the goods' transport.

Figure 2 presents a graphical representation of the line of argumentation for this theory.

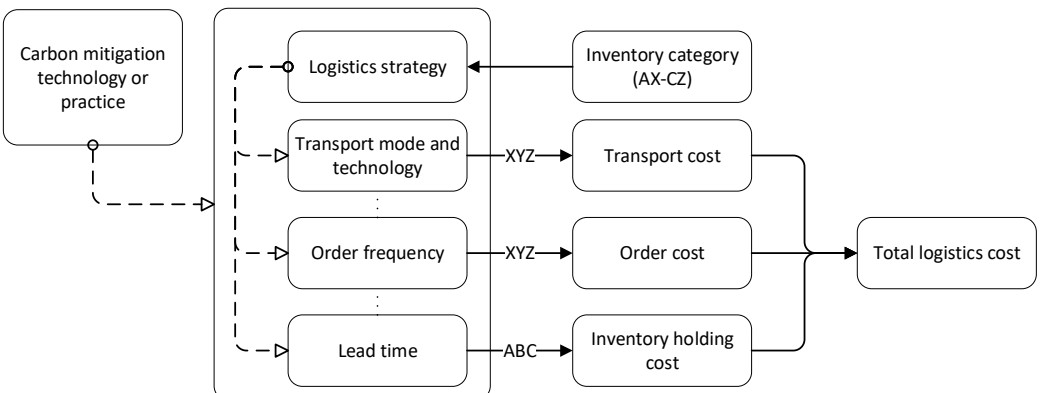

**Figure 2.** The impact of carbon mitigation practices on total logistics cost and the causal relationship with the inventory category.

This theory can be argued logically with different decarbonization measures, for example, vehicle selection. The truck class depends on several factors such as distance, speed, reliability, and flexibility. Items with a consistent demand (X items) may be transported in larger and slower trucks, while those with a volatile demand (Z items) require high-speed transportation in smaller trucks. A study by [13] found that abatement costs vary among various truck classes, which implies that the reduction of transportation emissions of X and Z items differ in this case. Additionally, there is evidence that the cost of reducing emissions for electric vehicles depends on their utilization [33], which suggests that the abatement costs for smaller, high-priority deliveries may be higher due to the need for low-utilization transport. Another example is the shift to lower-carbon modes of transportation, i.e., rail freight, which implies longer lead times. As elaborated above, these affect inventory holding costs. Thus, this measure is proposed to be more cost-effective for low-value goods (C) than for high-value goods (A), which implies different abatement costs for the transportation of A and Z items. The concept of differentiating abatement costs across inventory items is advocated by other researchers. For example, using different modes of transport

for individual items across the entire range of products was found to identify products that are relatively cost-effective to abate, as well as more expensive-to-abate products [37].

To sum up, this theory leads us to our final proposition: The abatement costs of one carbon mitigation practice or technology differ significantly between the ABC/XYZ category of the good to which it is applied; simply put, "the goods' inventory category impacts abatement costs of the goods' transportation". Table 1 presents a conclusive summary of the literature on this topic, which is used to argue for this proposition.

**Table 1.** Literature summary.

| Statement | References |
|---|---|
| Inventory management and its ABC/XYZ analysis are frequently applied in practice and well-researched | [16,34,35,37–41] |
| Inventory categories impact the selection of inventory strategies | [16,35] |
| Inventory strategies impact inventory cost | [16,35,43] |
| Inventory costs impact total logistics cost | [44,45] |
| Inventory strategies impact emissions | [46,47] |
| Implementing lower-carbon transportation technology, modes, or practices impacts total logistics cost | [13–15,20,33,48–50] |
| Selecting transport decarbonization measures on the product level can optimize total abatement cost | [37] |
| Transportation abatement costs differ across inventory categories | This study |

Although our proposition has been carefully developed, it needs to be thoroughly evaluated. In the following section, we present an initial evaluation of the theory applied to one decarbonization measure, i.e., combined road–rail transportation.

## 4. The Methodology for the Proposition Validation

In-depth validation of the theory across all potential decarbonization measures exceeds the scope of a single article; therefore, in this study, we have opted to test the theory by way of exemplification using one specific decarbonization measure. In the upcoming sections, we present the rationale for opting for combined road–rail transportation (CRRT) as a model decarbonization measure, evaluating it through a discrete-event simulation approach. Additionally, we will highlight the critical facets of CRRT that are considered during the simulation, along with the description of the two case studies that were used for testing the proposition.

### 4.1. The Selection of Combined Road–Rail Transport as an Exemplary Decarbonization Measure

For the proposition validation, we selected CRRT as the exemplary measure, which is substantiated by several key factors. First, the emphasis on shifting towards lower carbon modes of transport as a central pillar of decarbonization literature [7] underscores the relevance and significance of rail transportation. As inland waterway transportation faces challenges regarding reliability [51,52], which is an important decisive factor for transportation users [53], rail transportation is the more common shifting alternative [21]. Since only a limited number of manufacturing firms have direct access to the rail network, intermodal transportation, particularly combined road–rail transport (CRRT), is promoted by various logistics service providers (see, e.g., [54–56]). The demonstrated cost-effectiveness of CRRT in mitigating emissions in China [22] and Europe [57,58] affirms its practicality and environmental viability.

Despite the potential advantages of CRRT, it presents significant challenges for shippers due to increased lead times and the involvement of multiple stakeholders in this type of transportation [59–62]. Therefore, this measure presents greater evaluation complexity compared to others, such as changing drivetrain technology. While the logistics system stays consistent with a change in drive technology, CRRT planning necessitates significantly extended replenishment times and consideration of rail network delays. For cost

and emission assessments, it is necessary to consider two modes of transportation as well as load units and handling and storage procedures. This complexity creates challenges in conducting environmental and economic impact assessments. For these reasons, we opted to evaluate our proposition for CRRT, as it is a promising and hard-to-assess decarbonization measure. This selection aligns with the European Union's ambitious decarbonization goals, indicating that CRRT not only holds theoretical promise but also aligns with broader regional strategies for sustainable freight transport [63].

### 4.2. The Use of Discrete Event Simulation as the Evaluation Method

Because of the complexity of CRRT's impact assessment, simulation is a commonly applied methodology in researching intermodal transportation. Through simulation, researchers can gain a better understanding of mechanisms within the transportation system, reaching from the operations in transshipment hubs [64] to the utilization of transport corridors [65]. Simulation paradigms vary depending on the study's objective and encompass a multitude of possibilities, including agent-based [66], discrete-event [64], and system dynamics [67] approaches, as well as Monte Carlo simulation [49]. For the focal study, we developed a discrete-event simulation (DES) using Python and the Salabim library [68]. Within the simulation, the behavior of all elements and actors involved in CRRT, like consignees, shippers, hubs, trucks, trains, and load units, are modelled. We opted for DES because it is frequently applied to intermodal transportation [69–71] as well as the author's experience with DES.

### 4.3. The Case Studies Investigated for the Theory Validation

To validate our proposition, we use inbound shipment data from two Austrian industrial companies. The first case study is an Austrian electrical equipment manufacturer that provided us with nonfinancial data from its 2021 shipments over 11 months. Due to the attributes of the data, we cannot integrate the ABC part of the ABC/XYZ in this case study. For the validation of the ABC impact, we integrated a second case study, involving an Austrian cable manufacturer. The company provided us with shipment data from 2022, including the value of the goods.

For each case study, we combine all European inland road freight legs, including the final road leg for air or sea shipments, and road shipments as input for the DES. For each of the companies, we calculated costs and emissions from the status quo, i.e., road transportation, and compared it to two hypothetical shifting scenarios. Thereby, we were able to calculate hypothetical transportation abatement costs for each of the inventory categories.

Due to the numerous possibilities for planning, conducting, and controlling CRRT, we have made certain assumptions to ensure the simulation's effort is manageable. Therefore, the following constraints have been made:

- For competitive CRRT, the maximum number of transshipment operations during the main rail leg is set to one, i.e., no more than two rail services included in the main leg are allowed. The maximum allowed time for intermodal transportation is two days. To select the possible CRRT services, we search for the origin-destination pair on Routescanner.com (accessed on 11 September 2023) for each weekday and select the quickest connection that fulfils the requirements just mentioned. If there is no viable connection, we exclude the respective origin–destination pair from all scenarios. The rail services used are presented in the supplementary material.
- For pre- and post-haulage, we used the shortest possible routes from the suppliers to the origin terminal as well as from the destination terminal to the consignee. The distance and duration of these road haulages were acquired through the Openrouteservice.org (accessed on 11 September 2023) Distance API. We abstain from mentioning the precise addresses and distances due to confidentiality, but we can share that the average distances were 139 km and 169.5 km in the pre- and post-run for the first case, respectively, and 109.23 km and 193.82 km in the second case, respectively.

- The replenishment times differ between shipments from X, Y, and Z suppliers: X suppliers are given 7 calendar days replenishment time, Y shipments 5 days, and Z shipments 3 days. This reflects the predictability that is related to the different XYZ clusters. The replenishment time thereby determines how long before the planned arrival date the shipment is released by the supplier. For example, X suppliers are notified 7 calendar days in advance to send the shipment, either directly to the plant via road or to the first transshipment hub for intermediate storage.

- No further consolidation was considered. Shipment volumes and weights for intermodal shipments need to match the volumes and weights for the respective direct shipments that are given in the input data. It stands to reason that this is not the case with a real shift to rail as consolidation effects strengthen the business case for CRRT. However, we could not make any meaningful assumptions and would mix the evaluation with a second measure, which is why we decided against making any further assumptions on consolidation given the overall objective of the evaluation.

- To minimize cost, we aim to use 40-foot ISO containers. If the utilization of the 40-foot container is lower than 80% in terms of loading length, volume, or weight, we use 20-foot ISO containers instead. For competitive CRRT, each ILU utilization $u_{ILU}$ needs to be larger than or equal to 80%.

- Combining the former two assumptions led us to discuss how to handle shipments that have $1 < u_{ILU} < 1.8$. Therefore, we introduced two dispatching modes for those shipments:

    a.  In the first mode, we dispatch the whole shipment size to CRRT. This option constitutes the first shifting scenario, called "All ILUs on Rail" (ARA). In this scenario, the whole shipment, regardless of the shipment size, is dispatched to CRRT, meaning that one ILU is less utilized than 80%. This implies that some ILUs on rail are not well utilized, and the costs for renting, shipping, and handling the goods in this ILU are higher than for the better-utilized ones, but costs for direct road transport are obeyed. Before dispatching the ILUs via intermodal transportation, it is checked whether the ILU can reach the consignee on time with the available rail services. If this is not possible, the ILU is shipped directly by road transportation.

    b.  This implies more cost for a low-utilized road transport. In this scenario, a minimum utilization of 80% is necessary for each load unit to be shipped via CRRT. If this utilization is not reached, the ILU is scheduled for direct road transportation. We call this scenario "Highly utilized ILUs on Rail" (URA).

Therefore, including the base case, three scenarios are presented for each case study, which are visualized in Figure 3.

Table 2 presents a summary of the input data that was used for the scenarios. The difference between the total and considered shipments is based on constraints 1 and 2 of the enumeration.

The emission data that is used is a combination of dynamic and static data. For the transportation legs, we use the EcoTransIT World (ETW) emission calculator as the shipments reflect real-world transports that have been conducted. ETW offers a comprehensive calculation methodology that accounts for real-world movements of ships and flights and integrates up-to-date emission factors, emission quantification standards, and traffic networks [72]. To address ILUs in the hubs, we use a static value of 30 kg $CO_2$e per transshipment activity, as suggested by the GLEC framework [73]. Emission intensities of transshipment activities differ between hubs and equipment used, but no detailed data was available to us regarding terminal-specific emission factors.

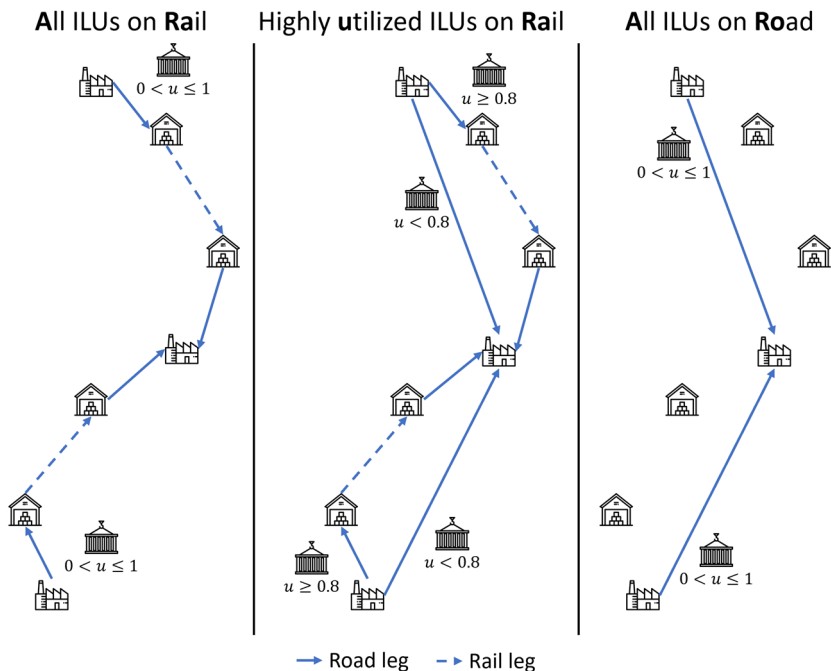

**Figure 3.** A schematic overview of the different dispatching mechanisms in the scenarios (from left to right the ARA, URA, and ARO scenarios).

**Table 2.** Descriptive statistics of the two cases under study.

| | Case 1 | | Case 2 | |
| | Total | Considered | Total | Considered |
| --- | --- | --- | --- | --- |
| Number of suppliers | 36 | 21 | 75 | 8 |
| Number of products | n.a. | n.a. | 359 | 10 |
| Number of shipments | 1215 | 275 | 1706 | 98 |
| Shipment date range | February–December 2021 | February–May 2021 | January–December 2022 | January–December 2022 |
| Transport GHG emissions | 548.51 t | 247.25 t | 305.60 t | 153.01 t |

*4.4. Total Logistics Cost of Combined Road–Rail Transportation*

Transportation costs generally comprise the cost components of each party involved in relocating goods [49]. Several studies have already defined the cost elements of intermodal transport and the comparison with unimodal transport. We use the cost functions defined in a recent case study on intermodal transportation [58] and detail them with other studies. The authors of [58] elaborate on the costs of intermodal transport services from a consignor's perspective and divide them into several parts. First, the pecuniary cost of transport *PC* includes the cost per kilometre and the cost of transshipment. Second, the monetary cost of transit time *TT* depends on the transport time, the value of the shipment, and the interest rate. Thirdly, the monetary cost of a delay *D* can be quantified, including fines and production downtime costs. Fourth, the cost of cargo loss *C* (i.e., damaged, expired, or stolen cargo) is a function of the value of the shipment, the fines, the production downtime costs, and the cost of reordering. Fifth, the study includes the cost of oversized cargo *OC*, which is highly dependent on the size of the cargo, and sixth, the social cost of transport *SCT*, the value of which depends on the method of quantification. Additional cost factors that have been considered in other studies are the expenses associated with storing a load unit at a terminal, the cost for the intermodal load units (ILU), incurred either as rent or depreciation, and the management and organization costs [74,75]. This study focuses on the perspective of the freight owner, which is why the total logistics costs are considered. Therefore, we have decomposed the aspects into six components that reflect the roles of all parties involved, including carriers, hub operators, railway operators, ILU owners, the organizing party, and the freight owner itself, including the transportation cost, order cost,

and inventory holding cost. Respectively, these components are the cost for transport operations *TC*, the cost for hub operations *HC*, the cost for the ILU rent or depreciation *ILUC*, the cost for production downtime *DC* and the cost for capital *CC*, as well as the overhead cost for organizing the CRRT *OC*. *TC* includes cost for pre-, main, and post-haulage, and *HC* includes the cost of the origin and destination hub. Figure 4 illustrates the occurrence of these components in the transport chain. As the freight owner is the principal of the transport, the costs are borne directly or indirectly by the manufacturing company. As we are missing data on ILU thefts or damages, we neglect these cost components in the focal study.

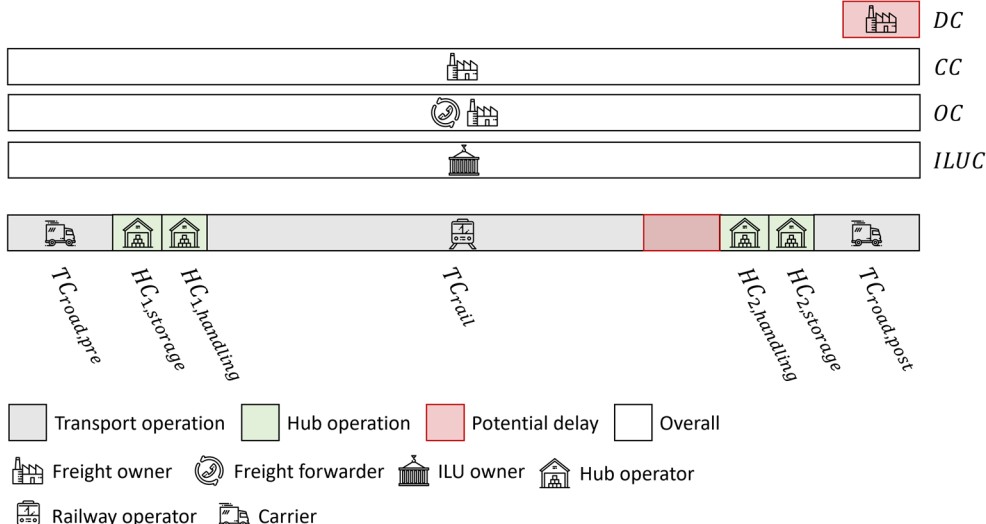

**Figure 4.** Cost structure of combined road–rail transportation from a freight owners' perspective (Icons from Freepik on Flaticon.com, accessed on 11 September 2023).

The cost data that was used in the simulation study is a combination of primary data and secondary data:

- $TC_{road}$: The input data from case study 1 were used to consider costs for direct truck transport. As a result, we developed a regressive function that depends on the truck utilization $u$ to determine the freight charges per tonne-kilometre $TC_{road,direct,tkm}$. Specifically,

$$TC_{road,direct,tkm}(u) = 0.0582 + \frac{0.0244}{u} \frac{\text{EUR}}{\text{tkm}}$$

  Additional information on this can be found in Appendix A.

- $TC_{rail}$: To obtain pricing information for the main leg, we consulted the sales team at a railway operator for data related to a sample CRRT service. They provided us with a cost of 800 EUR for a single 60-foot wagon travelling an on-rail distance of 822 km and having a capacity of three 20-foot containers, which are commonly referred to as 20-foot equivalent units (TEU). Based on this information, we estimate the cost of transporting one 20-foot ILU per kilometre as:

$$TC_{rail,km} = \frac{800\,\text{EUR}}{3\,\text{TEU}*822\,\text{km}} = 0.32 \frac{\text{EUR}}{\text{TEUkm}}$$

We are aware that rail transport pricing differs between operators, relations, and time, but regarding the efficiency of the research, we were not able to collect more detailed cost information empirically. Nevertheless, the 0.32 EUR/TEU used in this study is in the range of 0.46–1.35 EUR per forty-foot container provided by [75]. From this, we deduce that the exemplary price resulting from our research is representative of the European market.

- *HC*: For the hub cost, we use primary data from the involved terminals' homepages, if available. If not, the value of:

$$HC_{handling} = 48 \text{ EUR and } HC_{storage} = 0 \text{ EUR}$$

  is used in line with [75]. For gateway movements, we use:

$$HC_{handling} = 27 \text{ EUR and } HC_{storage} = 0 \text{ EUR,}$$

  in line with [75] as no other data is available to us.
- *ILUC*: The ILU rental fee per day was retrieved from an Austrian container rental company, compared with data from [76,77], and set to

$$ILUC = 7.5 \frac{\text{EUR}}{\text{day}}$$

- *OC*: Since actual road freight costs are included in the input data, the road *OC* is already included in the road freight charges.
- *DC*: As no data was available to us on the production delay, we initialized the simulation with:

$$DC = 500 \frac{\text{EUR}}{\text{hour}}$$

- *CC*: In the first case, we cannot integrate *CC* in our evaluation due to the abundance of the shipment monetary values. For the second case, we compute the costs of capital by utilizing a literature-based interest of 12% from [62].

*4.5. Distribution of Train Delay Times*

Freight train delays negatively impact the reliability of intermodal freight networks and are an important, but under-researched element of combined transport [78]. Freight trains are typically less reliable regarding the planned arrival times than trucks, which is a planning challenge for freight owners. Reasons for freight train arrival delays can be grouped into two sets of factors [79]. The first set describes a deviation of the departure time from shunting yards or hubs. The distribution of these deviations in Swedish shunting yards has been investigated by [80]. In their paper, the authors find that the log-normal and the gamma distribution best approximate the departure delays and early departures, respectively. The second set describes delays during the journey of the train. Network capacity utilization, weather conditions, construction sites, and many more factors determine the actual arrival time of trains. To model the delays of freight trains in our simulation, we thus cannot rely on the departure time delays, as the influence of the journey would be neglected. As creating models that predict the arrival time by combining both sets of factors is its own body of research [79], we concentrate on approximating the actual arrival delay using a probability distribution. Some work in this field has already been done. For example, the delay time of arriving passenger trains was approximated for Chinese railway stations through exponential distribution [81]. Although passenger train punctuality is much higher than freight train punctuality in Europe (see, e.g., [82,83]), we argue that the nature of the factors influencing arrival delays are similar, and thus, the type of distribution approximating passenger train delays also approximates freight train delays. Therefore, we use the exponential distribution with the probability density function:

$$f(d) = \lambda e^{\lambda d}, d \in \mathbb{N}$$

to model the delay *d* of a freight train. For parameterizing, we use the punctuality information of the Austrian Railways, reporting that the share of combined wagonload traffic with less than 30-min delay in Austria in 2022 was 52% [82]. By calculating the cumulative distribution function:

$$F(d) = 1 - e^{-\lambda d}$$

we approximate $\lambda \approx 0.02531$ so that $F(29) \approx 0.52$. This results in a mean delay of $E = \frac{1}{\lambda} = 39.51$ min and a median delay of $m = \frac{\ln(2)}{\lambda} = 27.39$ min. In the simulation, we incorporate the delay by adding it to the scheduled lead time $t_s$, thereby defining the actual lead time $t_a$ of a railway service as:

$$t_a = t_s + d$$

## 5. Validation Results

### 5.1. Results of the ABC/XYZ Analysis

In the first case study, the shipment data solely represents the supplier-level and not the product-level, which is why we conducted the XYZ analysis for the suppliers as suggested by [16]. The XYZ classification thresholds were set to $CV_{XY} = 0.3$ and to $CV_{YZ} = 0.6$. These values are in the range of the values presented by other literature, as elaborated in Table A1 in Appendix B. As values from the literature vary significantly, we established the thresholds to classify approximately 20% of suppliers as X, 50% as Y, and 30% as Z, as suggested by [16,38].

In the second case study, the shipment data contains weight and value data on the product-level. Thereby, each shipment represents the transport of a certain product from a supplier to the manufacturing plant. Due to the shipment data, we were able to conduct an ABC/XYZ analysis. Following the first case study, the XYZ thresholds were set to $CV_{XY} = 0.3$ and $CV_{YZ} = 0.6$. The ABC-thresholds were set to $s_{AB} = 0.8$ and $s_{BC} = 0.9$, whereby $s$ is the share of the cumulative value, which aligns with other articles, as shown in Table A1.

Appendix B presents figures plotting the distribution curves of the XYZ and ABC/XYZ analysis of both case studies.

### 5.2. Results of the Simulation Study

The simulation of each scenario assessing the 275 shipments of the first case took about 5 min, and the scenarios of the second case about 1.8 min. The high-level results of the simulation study are comprehensively visualized in Table 3, outlining the total logistics cost of the scenarios for each case study along with the GHG emissions.

**Table 3.** Total logistics cost and GHG emissions resulting from the three scenarios for each case study.

| | Case 1 | | Case 2 | |
|---|---|---|---|---|
| | *Total Logistics Cost* *EUR* | *GHG Emissions* *t $CO_2e$* | *Total Logistics Cost* *EUR* | *GHG Emissions* *t $CO_2e$* |
| ARO | 321,698.21 | 247.25 | 176,436.29 | 153.01 |
| ARA | 304,060.61 | 222.59 | 159,717.17 | 120.31 |
| URA | 308,854.52 | 224.32 | 166,241.26 | 131.04 |

In both cases, the ARA scenario is the best one from an economic and an environmental perspective. In the first case, costs are reduced by 5.8% and emissions by 9.97% when shifting all possible load units by rail. In the second case, this scenario mitigates cost by 9.48% and emissions by 21.37%. This is because the supplier structure in the second case is more consolidated. This can be seen from the statistics presented in Table 2, where indications for many small shipments are given, or in the ABC/XYZ visualization in Figure A3 in Appendix B, highlighting that, for example, a single product sourced by a single supplier accounts for more than 20% of the overall weight.

Interestingly, the URA scenarios result in slightly smaller cost and emission reductions, which was somehow surprising to us. For the first case, 3.99% cost and 9.27% emission reduction; and for the second case, 5.78% cost and 14.36% emission reduction are calculated.

Inventory holding costs were not included in the first case. Upon closer examination of the second case, these costs represent 1.23% of the total expenses in the ARO scenario, 8.25% in the ARA scenario, and 5.58% in the URA scenario. The differences between ARA

and URA are due to a larger number of shipments routed via intermodal transport in the ARA scenario; this leads to longer overall lead times, which results in higher inventory holding costs.

According to a research project on the EU Combined Transport, as reported in 2015, the cost breakdown for a CRRT of a semitrailer from Germany to Italy with 435 km on the main rail leg is as follows: 3% from the cost of the load unit, 15% from road pre-haulage, 19% from road post-haulage, 3% from the exporting terminal, 6% from the importing terminal, and 55% from the rail leg [84]. The simulation in this study presents higher shares for pre- and post-haulages and a lower share for the rail leg due to the long pre- and post-carriage distances covered in our simulations. Specifically, in the URA scenario of Case 1, pre-haulage and post-haulage expenditures account for 33% and 32% of the total costs, respectively. As a result, rail transport expenses are comparatively low, at only 17%. Nevertheless, the expenses for terminals and ILU closely resemble those from the empirical report, implying the authenticity and comparability of our results.

To validate the proposition made in the first part of this article, we compare the costs and emissions of the ARA and URA scenarios to the ARO scenario by calculating abatement costs on the ABC/XYZ level available. Results are presented in Table 4.

**Table 4.** Transportation abatement cost (EUR per mitigated t $CO_2e$) on the ABC/XYZ levels for the evaluated scenarios, with cost savings in green and additional costs in red.

| | | Case 1 | | | Case 2 | | |
|---|---|---|---|---|---|---|---|
| | | X | Y | Z | X | Y | Z |
| ARA | A | | | | −874.99 | 15.21 | −891.09 |
| | B | | | | −989.48 | −1169.40 | −1251.47 |
| | n.a. | −1510.92 | −552.97 | −1411.41 | | | |
| URA | A | | | | −849.64 | −63.24 | −892.56 |
| | B | | | | | | |
| | n.a. | −1514.00 | −513.90 | 123.79 | | | |

On the left side of the table, the results of Case 1 are presented in the bottom lines of the ARA and URA scenarios as financial data was not available (*n.a.*). On the right side, the results of Case 2 are presented in rows separated according to the ABC category. The absence of C items in the study can be explained using the constraints mentioned in Section 4.3. Due to the small size of C-shipments, it is not possible to utilize a 20-foot container to a sufficient level. For the URA scenario, even the B shipments are too small, as only highly utilized load units are routed intermodally.

In Table 4, we have colored the cells according to their cost-effectiveness, with the green cells representing strongly negative abatement costs—cost savings accompanying emission savings—and the red cells representing positive abatement costs—cost increases associated with emission reductions.

In Case 1, abatement costs for X- and Z- shipments are similar, while AC for Y-shipments are still negative, but three times higher. The investigation of the URA scenario for the first case highlights the differences between the dispatching strategies. While the X- and Y-shipments evaluate to similar values, costs for Z-shipments significantly differ from the ARA scenario. If the utilization of load units on the train is focused, the Z shipments cause additional costs for badly utilized fallback road transportation, which impedes higher costs and emissions at a level that raises the abatement cost to a positive value. Comparing the results of Case 1 with Case 2, abatement costs at similar levels are observed. Interestingly, the abatement costs for the Y-shipments of Case 2 are higher in ARA than in URA. Despite the otherwise relatively similar values, the comparison of the cases shows that a generalization of the absolute abatement costs is hardly possible. However, this also shows that the abatement costs do differ between the ABC/XYZ classes, which supports our initial proposition and approach presented in Section 2.

To validate the proposition, we examined the results of the simulation runs of the second case study in more detail. Therefore, we conducted Kruskal–Wallis tests for three samples (shipments of the ARA scenario, the URA scenario, and both scenarios combined) and found that there is a significant difference in the abatement costs for at least two inventory groups in all samples. Delving deeper into the pairwise comparisons of the inventory categories' abatement costs, we found that abatement costs between the inventory groups BX-AY, AZ-AY, and AX-AY significantly differ for the ARA and Total samples, and between the AX-AY and AZ-AY groups for the URA sample. The methods used for the tests are described in Appendix C. Results show that, at least across the XYZ inventory groups, abatement costs differ significantly. Investigating the boxplots in Figure A4 shows that abatement costs are highest for the Y-shipments and similar for X and Z shipments. The influence of the ABC dimension was significant only in the ARA and total sample, indicating that the influence of inventory holding costs exists, but is not decisive in most cases.

To summarize, the high-level results of the simulated cases and scenarios indicate that abatement costs do pertain to the ABC/XYZ category. Further statistical examination of the second case study showed that there is a significant difference in the abatement costs of some inventory categories. These findings let us conclude that we can partially accept the developed proposition.

## 6. Discussion, Implications, and Limitations

This paper's approach is twofold. In the first part, the authors provide a novel perspective on industrial logistics' decarbonization measures by combining a well-known inventory management tool with transport decarbonization measures. From these considerations, it is proposed that transport abatement costs of a specific measure differ when it is applied across the nine different ABC/XYZ inventory categories. The second part of this research validates this proposition for an exemplary mitigation measure, combined road–rail transportation, while presenting some limitations. The following sections will discuss the implications and limitations of our findings for research as well as practice.

### 6.1. Implications for Practitioners from Simulating Combined Road–Rail Transportation

In the CRRT simulation, the rail scenarios denoted as ARA and URA, in both evaluated cases, yield an overall reduction in cost and emissions, resulting in negative carbon emission abatement costs. Consequently, despite being suboptimal due to extended pre- and post-carriage distances, the cost-effectiveness of CRRT was demonstrated. However, a more nuanced examination of the results by the ABC/XYZ classification unveils that certain inventory categories exhibit higher costs and/or emissions compared to the reference scenario.

Other research in this field finds that overall total logistics costs can be reduced when the transportation mode is not selected holistically, but product-wise [37]. This, in turn, supports our findings that total logistics costs for one carbon mitigation measure differ across inventory categories, which, in turn, leads to different abatement costs. Further, the costs associated with inventory holding significantly contribute to the total logistics expenses, comprising 1.23% in the reference scenario and escalating to a range of 5.58% to 8.25% in the rail scenarios. Nevertheless, this does not offset the cost savings realized from the rail segment. These two observations underscore the necessity of a comprehensive assessment encompassing total logistics costs, rather than focusing exclusively on transportation costs when evaluating mitigation strategies, which is a common practice. While transportation costs constitute the predominant component for freight owners, they do not comprise the entirety. Moreover, making overarching statements about the competitiveness of CRRT is unwarranted, underscoring the imperative to diligently scrutinize each CRRT application individually due to the intricate nature of the intermodal system. Differing shares of the cost elements in our study from preceding studies [84] highlight the presence of numerous parameters that influence the costs of the intermodal system. Our results also lead to the finding that the widely referenced break-even distance definition is not

universally applicable to combined transport, which was also shown in other modelling studies [49].

In the ARA scenario, costs for direct road transportation are non-zero due to the incapacity of CRRT to deliver certain Z-shipments promptly to the consignee. This discrepancy arises from the stipulated replenishment time of three days for Z-shipments. In contrast, all Y-shipments, ordered five days in advance of their required delivery, can be feasibly transported via CRRT. This underscores the criticality of accurate production demand forecasting and the implementation of distinct dispatching strategies for varied inventory categories.

Despite the elevated abatement costs of AY inventory items in the ARA scenario of Case 2, the URA scenario registers higher total costs. This is attributed to the efficiencies achieved in B-shipments in the ARA scenario, which compensate for the inefficiencies in AY-shipments. This implies that, at least within Case 2, the 80% minimum utilization threshold may be set excessively high. If this level were lowered, the cost-effectiveness of CRRT could potentially be further enhanced.

Based on the insights derived from simulating CRRT, we posit that the marked disparities observed between cases, scenarios, and inventory categories underscore the imperative, first, to evaluate the total logistics costs holistically and, second, to optimize the dispatching strategy when transitioning to CRRT. Utilization levels and replenishment times emerge as pivotal determinants in total logistics costs when employing CRRT, potentially demarcating the boundary between cost savings and additional expenditures.

### 6.2. Implications for Practitioners from the Validation of the Approach

Regarding the proposition made, simulation results show that GHG abatement costs vary among different shipment categories (X, Y, and Z) in the URA scenario for Case 1, but less in the ARA scenario. For Case 2, a closer statistical investigation of the abatement costs on the shipment level showed that the abatement costs significantly differ between some inventory groups. Therefore, the more crucial ABC/XYZ dimension for the abatement costs was found to be the XYZ dimension.

The primary determinant in logistics is the logistics costs. When these costs vary among different alternatives, it implies that the likelihood of implementing certain alternatives is greater than others [14,15]. However, for decision-makers utilizing our developed method, the objective is not only to maximize economic savings but also to reduce emissions. To combine these two quantitative decision criteria, abatement costs can be calculated. We have demonstrated that the cost and emission impacts of shifting freight transport from road to combined road-rail transport differ depending on the transported goods. Briefly said, from a freight owners' perspective, GHG abatement costs for intermodal transportation vary, depending on the class of the inventory being transported. The most impactful variable, thereby, is the demand characteristics (order frequency and volume), described by the demand stability of the XYZ dimension. Therefore, we see that inventory holding costs, which are affected by the ABC dimension, are a less important criterion.

Nevertheless, our findings concurrently highlight the necessity for a nuanced assessment of the applicability of combined transport for different use cases to effectively exploit potential savings. The generalized assumption of break-even distances, dispatching strategies, or costs is not feasible, even within a single company. At the same time, the economic and environmental impact of a shift towards combined transport is difficult to evaluate in detail.

Since costs differ between some of the ABC/XYZ classes, and the ABC/XYZ analysis is a straightforward and well-known measure, this study provides practitioners with an efficient method for identifying potential promising use cases for combined transport. Even though the results differ between our simulation cases, a cost-effective reduction in emissions was demonstrated for the shift of X goods in every instance. For Y goods, the statistical tests showed a higher risk for positive abatement costs, and the Z goods varied across the case studies. This insight allows for minimal effort in identifying suitable goods

for combined transport and provides a starting point for CRRT newcomers, decreasing the entry barrier to the intermodal system. Therefore, this approach aids industrial enterprises in efficiently decarbonizing their logistics.

### 6.3. Limitations and Further Research Directions

As with every research article, there are limitations to this study which create opportunities for further research. The main limitation of this study is the requirement of using only one decarbonization measure to validate the proposed method. The primary objective for future research is to evaluate the proposition for additional decarbonization strategies and suggest the most effective measures for each of the nine inventory categories. A large volume of literature is available that recommends and categorizes decarbonization techniques (see, e.g., [20] for a comprehensive list), which can be a useful starting point for future investigations. For example, a frequently discussed measure is the use of trucks with alternative drives. It would be interesting to investigate effective application scenarios and the differences in abatement costs among ABC/XYZ items. To test for these differences, we suggest collecting real-world data from case studies that have implemented GHG mitigation measures for a broad range of inventory items or conducting simulation-based studies. The first approach may be criticized for its comparability, while the second may be questioned for its validity. Nevertheless, data availability is a crucial issue for all approaches, which is why it is recommended that research institutions collaborate closely with industrial companies to obtain realistic data.

Due to missing data, we neglect costs for damaged or stolen cargo. These have been modelled in other studies [58] and might harm CRRT's competitiveness. Although prior research indicates that CRRT is cost-competitive only when the pre- and post-haul distances are less than 40 km and the primary rail haul is no less than 750 km [75], the scenarios evaluated in this simulation are beneficial in terms of cost. Further, we have not included external costs in our study, as these do not currently have to be paid. However, a recent European study shows how dramatically different the results could be if external costs had to be paid in transportation [75].

Regarding CRRT, many different parameters can be adjusted, and many different replenishment, order, dispatching, and routing strategies can be followed. As intermodal transportation is a highly complex system, we were not able to fully test all parameters but had to make some assumptions regarding how CRRT could be conducted. Although these assumptions were discussed with other researchers and practitioners, future studies might assess the impact of a change in load unit compositions, route selection, and many more operational aspects of CRRT.

We were not able to validate the calculated figures for the mean and median of the train delay for Austria due to an absence of data. An analysis of train delays for intermodal freight services departing in Luxembourg resulted in $E = 74.35$ min and $m = 12$ min [79]. Another study found the standard deviation of freight train delays on one Swedish railway line to be $\sigma = 64.19$ min [85]. Assuming the underlying data is subject to an exponential distribution means that $E = \sigma$. Comparing our computed results, $E = 39.51$ min and $m = 27.39$ min, and with these empirical data, we see that our approximation produces smaller delay times. On the other side, an older study utilizes an average anticipated delay time of $D = 30$ min [62]. To address the ambivalence in the results, the suitability of the exponential distribution should be further researched in future studies. During the literature review on train delay times, we found that the majority of studies dealt with real-time delay prediction for passenger trains, but almost no studies presented high-level metrics for freight trains [86]. Nevertheless, for simulations like ours and logistics network planning for shippers and logistics service providers, further research on this topic could be helpful.

## 7. Conclusions

Throughout this study, we delineated and preliminarily validated the proposition that the ABC/XYZ analysis can be used to cost-effectively diversify industrial logistics decarbonization measures. In detail, we propose that abatement costs of a specific low-carbon practice or technology differ when applied to transport goods of different ABC/XYZ categories. For the preliminary validation, we used two simulation case studies, assessing the impact of shifting goods from road to combined road-rail transportation. Results show that abatement costs differ between ABC/XYZ categories in some scenarios, but we could not determine a common pattern across the two studies, except that X-shipments were cost-efficient in all scenarios. For a more detailed examination, we conducted a Kruskal–Wallis test to compare the abatement costs from shipments of the nine inventory categories of the second case study. Results show that there is a significant difference in the resulting abatement costs across some inventory categories, which suggests that our proposition can be accepted partially for combined road–rail transportation. Thereby, the inventories' XYZ dimension impacts abatement costs more significantly than its ABC dimension. Nevertheless, the results of our tests refer to only two case studies, which is why more research is necessary to assess additional scenarios, supplier–consignee relationships, and industrial sectors. For researchers, this study provides a promising foundation for investigating abatement costs, which are a critical component of efforts to decarbonize transportation. For practitioners, the findings of this study indicate that the makeup of a company's inventory has a significant bearing on the costs of transporting goods in an ecologically responsible manner. This is vital for making knowledgeable choices in industrial logistics and harmonizing economic effectiveness with environmental responsibility.

**Supplementary Materials:** The following supporting information can be downloaded at: https://www.mdpi.com/article/10.3390/app132212277/s1.

**Author Contributions:** Conceptualization, P.M.-B.; methodology, P.M.-B.; software, P.M.-B. and M.H.; validation, P.M.-B. and M.H.; writing—original draft preparation, P.M.-B.; writing—review and editing, M.W. and P.M.-B.; visualization, P.M.-B.; supervision, M.W.; project administration, M.W. All authors have read and agreed to the published version of the manuscript.

**Funding:** This research belongs to the project "SME 5.0—A Strategic Roadmap Towards the Next Level of Intelligent, Sustainable and Human-Centred SMEs" (funded in the European Union's Horizon MSCA 2021 programme under the Marie Skłodowska-Curie grant agreement No. 10108648).

**Institutional Review Board Statement:** Not applicable.

**Informed Consent Statement:** Not applicable.

**Data Availability Statement:** Data is contained within the Supplementary Material.

**Conflicts of Interest:** The authors declare no conflict of interest.

## Appendix A

To analyze the freight charges, we used visualizations to examine the input data. Our investigation revealed a clear non-linear correlation between truck utilization and freight charges per tonne-kilometre. Using scikit-learn, we created a Python script to fit our data to the regressive function $y(x) = a + \frac{b}{x}$. The code we used is accessible online at https://github.com/MUL-Chair-of-Industrial-Logistics/simple_regression (accessed on 11 September 2023) and outputs $a = 0.05816489075412676$ and $b = 0.02437902515302302$. The left subplot of visualizes the regression results.

Although we anticipated an association between transportation distance and freight charges per tonne-kilometre, we failed to identify a function that accurately models this relationship with acceptable residuals. The data point distribution is displayed in the right plot of Figure A1. This could be because the transportation data available to us primarily included long-distance shipments.

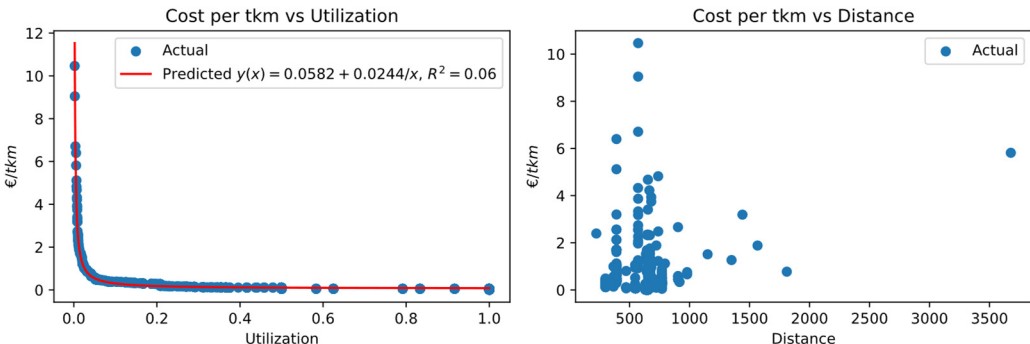

**Figure A1.** Graphical representation of the freight charges analysis.

## Appendix B

The left plot of Figure A2 displays the characteristic curve of the XYZ-analysis plotted against the number of suppliers, while the right plot shows it against the cumulative weight.

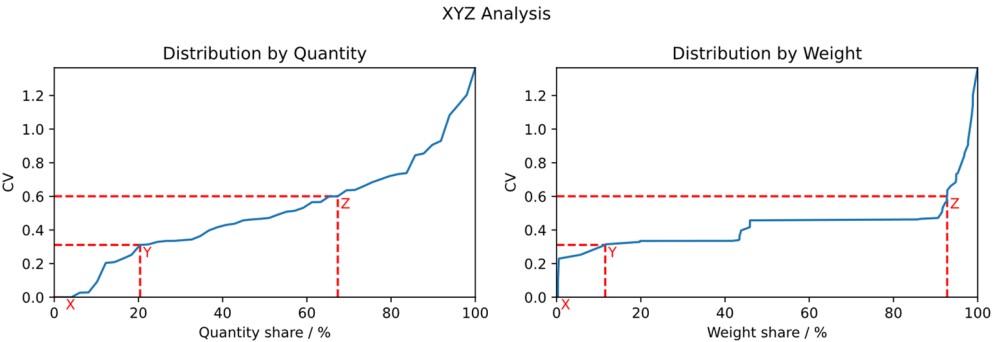

**Figure A2.** The characteristic curve of the XYZ analysis plotted against the number of suppliers (**left**) and the cumulative weight (**right**) for the first case study.

In Figure A3, the results of the ABC/XYZ analysis for the second case study are visualized.

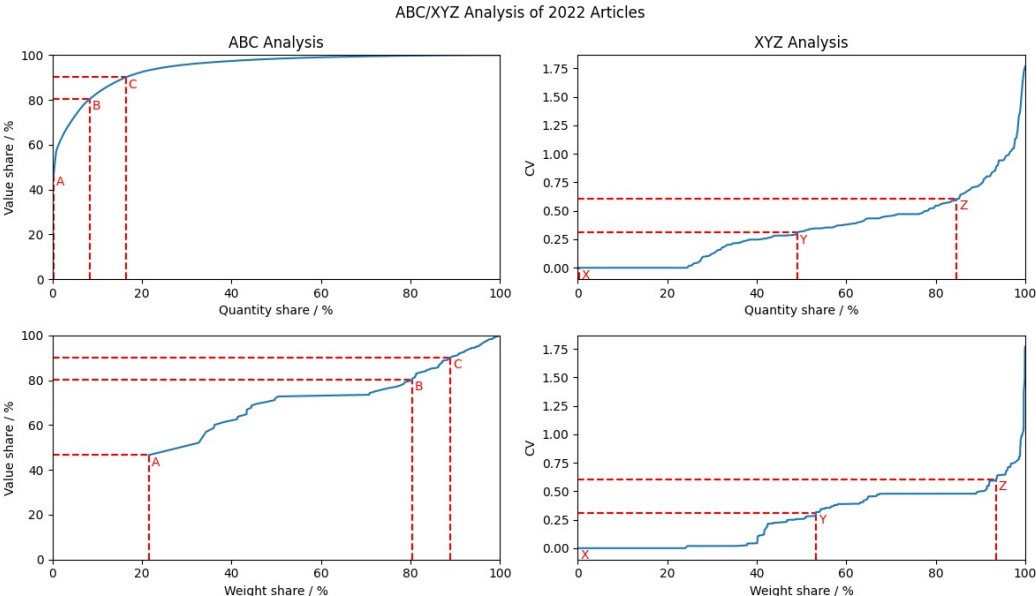

**Figure A3.** The characteristic curves of the ABC analysis (**left**) and the XYZ analysis (**right**) are plotted against the number of products (**top**) and the cumulative weight (**bottom**) for the second case study.

Table A1 presents the ABC/XYZ thresholds of this study in comparison to other references.

**Table A1.** Comparison of ABC/XYZ thresholds from the literature.

| Reference | $s_{AB}$ | $s_{BC}$ | $CV_{XY}$ | $CV_{YZ}$ |
|---|---|---|---|---|
| *This study* | 0.8 | 0.9 | 0.3 | 0.6 |
| [16] | 0.7–0.8 | 0.9–0.95 | 0.1 | 0.25 |
| [38] | 0.6–0.8 | 0.85–0.95 | | |
| [87] | 0.8 | 0.95 | 0.5 | 1 |

**Appendix C**

To test the significance of the results, we calculated the abatement costs of each shipment shipped in the ARA and URA scenarios in the second case study. Then, we calculated the Kolmogorov–Smirnov and Shapiro–Wilk tests to test the abatement costs for normality using SPSS. We use an alpha value of $\alpha = 0.05$, which leads to a rejection of the hypothesis that the distribution of abatement costs is similar to a normal distribution. The results of the test for the shipments of the URA scenario, the ARA scenario, and both shipment data combined are presented in Table A2.

**Table A2.** Normality tests.

| | Kolmogorov–Smirnov [a] | | | Shapiro–Wilk | | |
|---|---|---|---|---|---|---|
| | Statistics | df | Sig | Statistics | df | Sig |
| abatement_cost_ara | 0.209 | 55 | <0.001 | 0.866 | 55 | <0.001 |
| abatement_cost_ura | 0.267 | 32 | <0.001 | 0.776 | 32 | <0.001 |
| abatement_cost_total | 0.229 | 87 | <0.001 | 0.852 | 87 | <0.001 |

[a] Significance correction according to Lilliefors.

Due to the non-normality of the sample, we selected the Kruskal-Wallis – test for further investigation [88]. The test determines if all populations of one sample are identical [89]. In the focal case, the null hypothesis is that abatement costs are identical across the inventory categories, which implies that the alternative hypothesis is that abatement costs differ across inventory categories. Again, the alpha value was set to $\alpha = 0.05$. For all three samples (ARA shipments, URA shipments, and both shipments combined), the null hypothesis was rejected, and the alternative hypothesis was accepted, meaning that a statistically significant difference was found between at least two groups in all samples. Test results are displayed in Table A3.

**Table A3.** Kruskal–Wallis to test the null hypothesis.

| | Kruskal–Wallis | | | |
|---|---|---|---|---|
| | Chi-Square | df | Sig | H |
| ARA | 38.429 | 5 | <0.001 | reject |
| URA | 19.176 | 2 | <0.001 | reject |
| Total | 58.644 | 5 | <0.001 | reject |

The pairwise group comparisons of the ARA and Total sample show significant differences ($sig < \alpha$) between the groups BX-AY, AZ-AY, and AX-AY. The group comparison of the URA scenario shows significant differences between the groups AX-AY and AZ-AY. These results are visualized in the boxplots shown in Figure A4, whereby SPSS visualizes outliers by circles (difference to 1st or 3rd quartile > 1.5 * interquartile range) and extreme outliers by asterisk (difference to 1st or 3rd quartile > 3 * interquartile range).

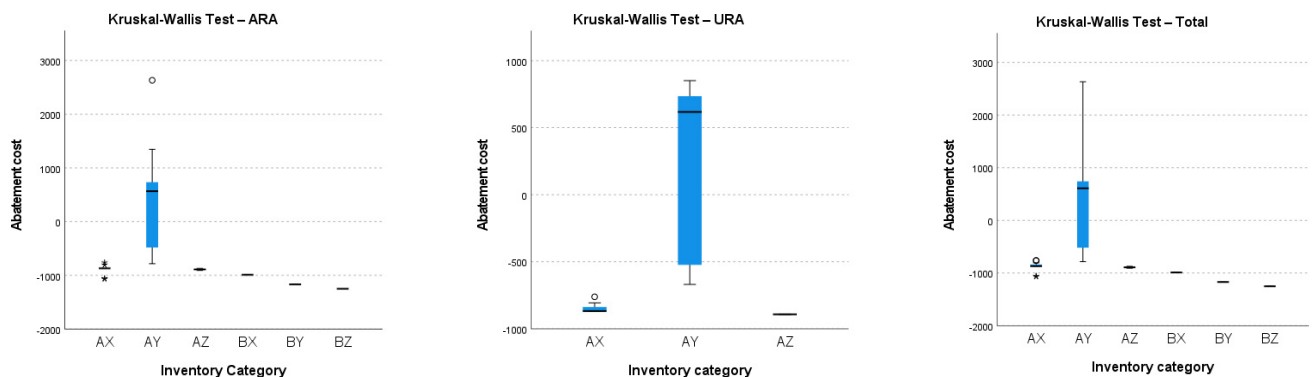

**Figure A4.** Boxplots of the abatement cost results for the ARA, URA and Total samples (left to right).

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
