# Peer review of "A Novel Approach to Identify Industrial Logistics Decarbonization Opportunities: Method Development and Preliminary Validation"

_applsci, doi:10.3390/app132212277_

Round 1
Reviewer 1 Report
Comments and Suggestions for Authors
- The construction of the conceptual basis of the work is adequate.
- It is necessary to characterize better the research gap, explaining its contribution comparatively to other existing works. In presenting the research problem, the authors must indicate, comparatively (through a summary table), how the work proposal represents an advance about the other works discussed in the literature review.
- The research method is fragilized by the limitation caused by the data obtained for analysis.
- It is necessary to present the implications of the work for the real world to enable the assessment of its applicability.
Comments on the Quality of English Language
The article needs only a few improvements in the text.
It is necessary to correct the presentation of citations in the text: errors of this nature appear in lines 210 (p. 5), 279 (p. 7), and 380 (p. 9).
Reviewer 2 Report
Comments and Suggestions for Authors
The authors of the article attempt in their research to prove somewhat controversial in my opinion thesis that "the most efficient measures to reduce GHG emissions, in terms of abatement costs, vary among different ABC/XYZ inventory categories." However, it should be noted that:
1. the description justifying the selection of means of transportation to product groups grouped according to the XYZ classification, and indeed groups resulting from the ABC classification, is clearly insufficient. While the rationale for selecting the type of truck for products with fixed and variable sales characteristics I can justify, the impact of the value of inventory on the selection of the means of transport is already a big surprise to me. This section needs a solid follow-up and justification.
2 Figure 1 requires a detailed discussion. It is challenging to consider that the visualization of the theory formulated as a hypothesis was presented on it.
3. The description of the methodology needs improvement. Why do the authors first elaborate on selecting different trucks into freight groups according to the XYZ classification and then suddenly consider only the road-rail connection? The authors should provide a detailed description of the research procedure. Especially in the article's title, there is a phrase concerning "novel approach" and "method development."
4 The new approach proposed by the authors concerns using ABC/XYZ classification to identify industrial logistics decarbonization opportunities. Unfortunately, the results presented only apply to the XYZ classification. Instead, the entire potential substantive contribution focuses on using a combination of ABC/XYZ classifications. Unfortunately, this is not presented in the article.
I suggest that the authors rethink the hypothesis and verify the results obtained from their research work. The article needs a significant revision.
Author Response
Please see the attachement

Reviewer 3 Report
Comments and Suggestions for Authors
I kindly ask the authors to correct the indicated editing errors and answer the questions below:
Line 159 - 161 - "Inventories frequently constitute a considerable portion of the total assets on the balance sheet of manufacturing companies – a 15-20% share is not uncommon" - please indicate the source of these data.
line 210 - "Combining this with the findings of Section Error! Reference source not found..." - please correct
Line 277 - 279 - "Furthermore, delays in the train network are much more common and impactful than on the 278 road – for a detailed discussion on this see Section Error! Reference source not found..." - please correct
Line 380 - "The left plot of Error! Reference source not found..." - please correct
Lines 400-401 - "The rail services used are pre-400 sent in Error! Reference source not found..." - please correct
Please expand and rebuild the Conclusions section in such a way that the information contained therein corresponds to the results of the analyzes presented by the authors in the article.
In line 58, the authors proposed a research hypothesis: "different transportation requirements come different costs for 58 lower-carbon transportation". However, in line 604 they provide information about the limitation: "The main limitation in the simulation study regarding the validation of the hypothesis is the absence of financial data for the shipments". Therefore, I am asking for a precise answer as to what was the point of formulating the main research hypothesis of the article if the authors did not have data to verify it?
Author Response
Please see the attachement

Reviewer 4 Report
Comments and Suggestions for Authors
This study explores the application of inventory control tools in the context of decarbonizing transportation. The use of ABC/XYZ analysis to categorize items and its impact on cost-effectiveness in decarbonization strategies is innovative. The discrete-event simulation provides valuable insights, highlighting differing abatement costs for various shipment types. Overall, it offers a promising framework for efficient decarbonization decision-making in industrial logistics, with practical implications for addressing climate change.
Author Response
Dear reviewer,
thank you very much for the positive feedback on our article!
Best regards,
the authors
Reviewer 5 Report
Comments and Suggestions for Authors
The authors present a new approach to identifying decarbonisation opportunities in industrial logistics and its preliminary validation. It is well structured and supported by a large number of references, but some considerations should be made:
Keywords should not be bolded.
The paper presents the equations inserted in the body of the text. I think the document would be more visible if the equations were inserted according to the "Mathematical Components" formatting of the template.
The text contains some unidentified acronyms (e.g., GHG, ROI, TEU, FEU). Describe them the first time they appear in the text.
Check that the reference [10] on line 189 and following is correct. Is it [40]?
There are several referencing errors in the document: lines 210, 279, 380 and 401.
Reference [59] is not mentioned in the text.
The data presented in the left graph of Figure 3 is not discernible. Redefine the range of the YY axis.
The appendices should be referred to in the order in which they are referred to in the body of the text. On line 445, the second appendix "B" is mentioned first. Appendix "A" is not referred to until line 512. Order your referencing.
References 9 and 18. Some references are quite old. Can they be replaced with more recent references?
Can the conclusions presented by [20] (point 3.3) be transposed to the European reality? Is the quality of the vehicles the same? Are their emissions lower, the same or higher? This analysis must be carried out in order to adapt it to the European reality.
The conclusions presented do not show effective results from the adoption of this new model proposal. The authors should consolidate the conclusions by pointing out the concrete results obtained and develop them in the near future to make them credible.
Author Response
Please see the attachement

Round 2
Reviewer 1 Report
Comments and Suggestions for Authors
After improving the article, I think it's nearly ready for publication. I suggest conducting a final review for spelling, grammar, and punctuation mistakes before completing the manuscript.
Comments on the Quality of English LanguageI suggest reviewing the text one last time before presenting the final version of the article.
Author Response
Dear reviewer,
thank you for your constructive feedback. We have thoroughly checked the manuscript for punctuation, grammar and spelling errors and thereby increased its quality significantly.
We hope the manuscript now meets your expectations and the typical academic language and thank you very much for your time and effort!
Yours sincerely,
the authors
Reviewer 2 Report
Comments and Suggestions for Authors
Many thanks to the authors for the changes made to the article. They had a positive impact on the quality of the material presented. Unfortunately, several issues still need to be corrected and supplemented. I have included the most essential comments below. They are critical to be able to allow the article for publication.
1. Line 293: Why are low-cost materials labeled Z and high-cost materials labeled A? Which classification are the authors referring to in this case?
2. Please indicate the literature source based on which the values for CV in the XYZ classification and the values of S parameters for the ABC classification were established. Let me remind you that the XYZ classification is objective, and the CV parameters are derived from statistical laws. On the other hand, the ABC classification derives from the Pareto principle. Therefore, the SAB parameter should oscillate around the magnitude of 80%. Violation of these rules requires an explanation.
3. How come there is no group C in case 2? Did the authors of the article eliminate this group from the analysis? If so, this requires a detailed justification. I would like to remind you that the ABC classification applies to the entire set of analyzed indexes, so there is no way that the analytical procedure could have resulted in the absence of group C materials.
4. Given the above comments, I remain unconvinced by the proposed approach for relating the ABC/XYZ classification results to selecting an appropriate transportation system. In my opinion, the use of ABC classification for vehicle selection still needs to be justified. Case 2 is interesting, but its interpretation needs to be supported by adequate factual justification, especially regarding the results for ABC classes.
5. Since this is a verification of a new approach, the discussion should include references to the results of studies presented by other authors cited in the article's theoretical part.
Author Response
Please see attachement.

Reviewer 3 Report
Comments and Suggestions for Authors
All corrections introduced by the authors significantly improved the quality of the article. In my opinion, the material is suitable for publication in a journal.
Author Response
Dear reviewer,
Thank you very much for your positive feedback on our article!
Yours faithfully,
The authors